# Iron Bioavailability in the Extracellular Environment Is More Relevant Than the Intracellular One in Viability and Gene Expression: A Lesson from Oligodendroglioma Cells

**DOI:** 10.3390/biomedicines11112940

**Published:** 2023-10-31

**Authors:** Stefania Braidotti, Debora Curci, Daniele Zampieri, Cesare Covino, Davide Zanon, Natalia Maximova, Roberto Sala

**Affiliations:** 1Department of Pediatrics, Institute for Maternal and Child Health-IRCCS Burlo Garofolo, 34137 Trieste, Italy; stefania.braidotti@burlo.trieste.it; 2Advanced Translational Diagnostic Laboratory, Institute for Maternal and Child Health-IRCCS Burlo Garofolo, 34137 Trieste, Italy; debora.curci@burlo.trieste.it; 3Department of Chemical and Pharmaceutical Sciences, University of Trieste, 34127 Trieste, Italy; dzampieri@units.it; 4Advanced Light and Electron Microscopy Imaging Centre (ALEMBIC), IRCCS San Raffaele Scientific Institute, 20132 Milan, Italy; covino.cesare@hsr.it; 5Pharmacy and Clinical Pharmacology Department, Institute for Maternal and Child Health-IRCCS Burlo Garofolo, 34137 Trieste, Italy; davide.zanon@burlo.trieste.it; 6Department of Medicine and Surgery, University of Parma, 43121 Parma, Italy; roberto.sala@unipr.it

**Keywords:** oligodendroglioma, iron, pediatric, brain tumors

## Abstract

Oligodendroglioma (OG) is a brain tumor that contributes to <1% of brain tumor diagnoses in the pediatric population. Unfortunately, pediatric OG remains without definitive molecular characteristics to aid in diagnosis, and little is known about the tumor microenvironment. Tumor cells’ metabolism and proliferation rate are generally higher than those of healthy cells, so their iron demand is also significantly higher. This consideration underlines the great importance of iron for tumor development and progression. In this context, this study aims to evaluate the effect of iron in a cellular in vitro model of human oligodendroglioma brain tumor. Cell morphology, the effect of siderotic medium on cell growth, iron uptake, and the expression of iron-metabolism-related genes were evaluated via optic microscopy, ICP-MS, confocal microscopy, and real-time PCR, respectively. This study underlines the great importance of iron for tumor development and progression and also the possibility of reducing the available iron concentration to determine an antiproliferative effect on OG. Therefore, every attempt can be promising to defeat OG for which there are currently no long-term curative therapies.

## 1. Introduction

Brain tumors are the most common tumor and the first cause of tumor death in children, accounting for 25% of childhood cancers, second only to leukemia in frequency [1,2]. Brain tumors can be classified based on origin, glial or not, and represent a dramatic health problem due to their high morbidity and mortality in all ages [3].

Oligodendroglioma (OG, ORPHA: 251627, OMIM: 137800, 616568) is a rare subtype of glioma originating from oligodendrocytes, with a limited response to the current standard of treatment [4]. It contributes to <1% of brain tumor diagnoses in the pediatric population, where it is much less prevalent than in adults. Knowledge about pediatric OG is limited, with few studies examining different treatment approaches and outcomes [4]. The overall 5-year survival for grade II OG in the pediatric population is estimated to be 90%, while for grade III anaplastic OG, it is only 53% [5]. The standard of care for pediatric OG includes surgical resection, radiation, and a regimen of chemotherapy [6].

Given their rarity, the literature is sparse. Current knowledge states that pediatric OGs are often molecularly distinct from their adult counterparts and often lack the 1p/19q co-deletion or *IDH* mutation. However, little is known about the pediatric tumor microenvironment.

Tumor cells’ metabolism and proliferation rate are generally higher than healthy cells, so their iron demand is also significantly increased. Several epidemiological studies have investigated the association between body iron levels and cancer, supporting a positive association in adults [7]. The central nervous system is not directly in contact with the plasma iron pool because it resides behind the blood–brain barrier (BBB). Like many molecules, the entry of iron is tightly regulated by the BBB, and specific transport mechanisms are required to transfer iron into the brain tissue [8,9]. The BBB is of great significance in maintaining the stability of several factors in the internal environment of brain tissue and in preventing harmful substances in the blood from entering the brain tissue. It also controls iron transport from the bloodstream to the brain parenchyma, providing some resistance to systemic iron toxicity [10]. Different cerebral cells acquire iron through different pathways, which involve several iron transporters.

In addition, it is known that cells receive iron support for growth and proliferation, especially cancer cells [11,12]. Once iron has been taken up from the extracellular space, its presence in the cytosol can cause oxidative stress until the iron is safely stored within ferritin [13]. Evidence has shown that iron metabolism is closely related to glial cell tumorigenesis, tumor progression, and tumor resistance in the microenvironment, although key iron metabolism-related genes are unclear [8,14,15]. Since tumor cells strongly depend on iron for their growth/proliferation, they are more sensitive to iron depletion than normal cells [16]. Inducing iron depletion could have a therapeutic advantage to manage siderosis. Iron chelators have long been known for their effect on iron reduction and their antiproliferative effect [17,18]. It is unclear whether tumor resistance to conventional therapies occurs because of increased permeability of the BBB due to conventional pharmacological therapies or radiotherapy, allowing the massive entry of iron without control. This imbalance in cancer mainly manifests as increased iron metabolism, iron affinity, iron input, and inhibition of its output, resulting in iron accumulation. Elevated serum ferritin levels are observed in several types of cancer; however, little is known about the association between ferritin and glioma [19]. Only a few studies have assessed the ferritin levels in gliomas using routine laboratory diagnostics. However, the biological background of glioma-associated ferritinemia remains not well elucidated.

This consideration underlines the great importance of iron for tumor development and progression and also the possibility of reducing the available iron concentration to have an antiproliferative effect on OG. This study is part of a larger research project that aims to study iron permeability across the BBB as a trigger for the onset of brain tumors in pediatric patients with long-term survival after acute lymphoblastic leukemia treatments [20,21,22]. In this work, we focused on establishing an in vitro model of human oligodendroglioma brain tumors and characterizing cells undergoing siderotic culture conditions.

## 2. Materials and Methods

### 2.1. Cell Lines and Culture Conditions

The study was performed on immortalized human oligodendroglioma cell line (HOG) purchased from Sigma-Aldrich (SCC163, Merck KGaA, Darmstadt, Germany) [23,24]. We developed a siderotic HOG cell line model by culturing HOG cells for three months in low-glucose Dulbecco’s modified Eagle’s medium (LG-DMEM) (ECM0749, EuroClone SpA, Milan, Italy) in which the concentration of ferric nitrate was 0.25 µM, supplemented with 1% fetal bovine serum (10270106 FBS, Origin Brazil, Gibco- Thermo Fisher Scientific, Monza, Italy) 2 mM glutamine (35050, GlutaMAX Gibco-Thermo Fisher Scientific, Monza, Italy), and 1% penicillin–streptomycin solution (A8943, AppliChem, Darmstadt, Germany) enriched with 100 µM of ferric citrate (F3388, Merck KGaA, Darmstadt, Germany). We cultured HOG cells, our control cells, in complete culture medium, without any ferric citrate supplement (the complete medium’s iron concentration did not exceed 0.75 µM). Cells were kept in cell culture flasks (#833911.002 Sarstedt, Nümbrecht, Germany) at 37 °C in a humidified atmosphere with 5% CO_2_. Cell culture experiments were conducted at Parma University.

### 2.2. Viability Measurements (Alamar Blue Assay)

To analyze the cells’ growth kinetics, cells were plated in 96-well plates (#833924 Sarstedt, Nümbrecht, Germany) (5000 cells/well for both siderotic and control cells) and then incubated in complete culture medium, supplemented or not with 10 µM of ferric citrate. After the indicated time, the growth medium was replaced with 120 µL of resazurin solution in DMEM (10 µg/mL) for one hour at 37 °C. Relative fluorescent units (RFUs) emitted by resorufin were acquired from 100 µL samples using an EnSpire^®^ Multimode Plate Reader (Perkin Elmer, Waltham, MA, USA) (λ ex/λ em. 540/585 nm).

Quadrupled wells were measured for each time point. The experimental setting was repeated at least three times. Cell growth kinetics were determined starting from day 0; analyses at days +1, +2, +4, +7, +9, +11, +14, +16, +18, +21, and +23 were considered.

### 2.3. Sample Preparation for ICP-MS Analysis in Cell Lines

Control and siderotic HOG cells were seeded onto 24-well plates (#833922-Sarstedt, Nümbrecht, Germany) at a density of 40,000 cells/cm^2^. The day after, the medium was substituted with a growth medium based on DMEM supplemented or not with either 25 µM or 100 µM of ferric citrate. After the incubation, cells were washed 5 times with PBS and detached using trypsin solution. After trypsin neutralization with DMEM + 10% FBS, the cell suspension was centrifuged for 5 min at 200× *g*; thereafter, the pellets were fixed in 100 µL of paraformaldehyde solution, 4% in PBS, for 10 min at 4 °C.

### 2.4. Quantification of Iron (Fe) in Cell Lysates

To the samples, we added 50 µL of 69% nitric acid (VWR International S.r.l., Milan, Italy) and diluted to a final volume of 5 mL with Milli-Q water (Millipore purification pack system, Millipore, MA, USA). Cell samples were mineralized with 200 µL of nitric acid (69%) and 50 µL of hydrogen peroxide 30% (Sigma-Aldrich, Milan, Italy). Subsequently, cells were sonicated for 15 min and diluted to 5 mL with Milli-Q water.

An ICP-MS Nexion 350X with an ESI autosampler (Perkin Elmer, Waltham, Massachusetts, MA, USA) was used to determine the total Fe concentrations in cell lysates derived from the mineralization. Five-point standard curves obtained via the dilution of Fe standard solutions (10000 µg/L) for ICP analysis (Sigma-Aldrich, Milan, Italy) were used for ICP-MS measurements (range 0.01–100 μg/L; ion mass: 54 and 57 u.m.a. for Fe). Laboratory-fortified blanks and laboratory-fortified samples (1 µg/L and 10 µg/L) analyzed before and after the sample solutions were used to assess the matrix effects.

The recovery for Fe was always >95%. The limit of detection was 0.1 µg/L for Fe, and the precision of the measurement repeatability (relative standard deviation (RSD) %) for the analysis was <3%.

### 2.5. Confocal Laser Scanning Microscope Analysis

Confocal fluorescence microscopy was accomplished at the Confocal Microscopy Laboratory DIMEC, University of Parma.

Cells were seeded into 35 mm cell view culture dishes with glass bottoms (627860 Greiner Bio-One S.r.l., Milan, Italy) in complete medium, supplemented or not with ferric citrate. Before the analysis, cells were rinsed twice in PBS and incubated for one hour in a complete medium, supplemented or not with 10 µM of ferric citrate, in the presence of 5 µM of Rhonox-1, and 1.6 µM of LysoTracker^®^ Yellow HCK-123. For nuclei counterstaining, 5 µg/mL of Hoechst 33-258 was added to all the tested conditions during the last 20 min of incubation.

Dishes were inserted into a heated chamber (Oko UNO Stage Inc Set Premix Inv K-Frame. 158006102-Okolab https://okolab4microsystems.com/, accessed on 30 July 2023) warmed to 37 °C with 5% CO_2_. All images were collected with a Leica TCS SP5 Laser Scanning Confocal microscope (Leica Microsystems, Wetzlar, Germany), equipped with a 405 nm (diode) laser line (LL) and an HCX PL APO λ blue 63X (NA 1.4) oil objective, and LAS X 4.3.0.24308 microscope software. Single confocal sections adopted independent configuration settings to avoid any possible crosstalk during acquisition. The emission of the Rhonox-1 signal excited at 540 nm LL was collected with a spectral detection ranging from 548 to 647 nm. Hoechst 33-258 was excited at 405 nm LL, with a spectral detection ranging from 420 nm to 486 nm.

The co-localization of Fe^2+^ and lysosomes was evaluated using the following setting: Hoechst 33-258 was excited at 405 nm LL, with a spectral detection ranging from 410 nm to 459 nm. LysoTracker^®^ Yellow HCK-123 was excited at 485 nm, and emission was collected with a spectral detection ranging from 571 to 705 nm. The Rhonox-1 probe was excited at 543 nm LL, and emission was collected with a spectral detection ranging from 552 nm to 599 nm. Fluorescence intensity of the single cells was acquired, and images were analyzed with ImageJ software ver. 1.54g. Thereafter, we calculated the corrected total cell fluorescence (CTCF) for each cell with the following formula:CTCF = integrated density − (area of selected cell × mean fluorescence of background readings).

Data of intracellular Fe^2+^ were expressed in each cell as CTCF/area of the selected cell. CTCF was measured in 69 cells of the control group (HOG LG-DMEM), 73 cells of HOG LG DMEM+ 10 µM Fe-citrate, and 57 cells of the HOG100 LG-DMEM + 10 µM of Fe-citrate population.

Additional analyses were carried out in ALEMBIC, an advanced microscopy laboratory established by IRCCS Ospedale San Raffaele and Università Vita-Salute San Raffaele (Appendix A).

### 2.6. Gene Expression by Real-Time PCR Analysis

Total RNA was extracted from HOG, HOG 100 µM cultured with iron citrate, and HOG 100 µM cultured without iron citrate, using a Monach^®^ Total RNA Miniprep Kit (#T2010S, New England BioLabs. Inc, Ipswich, MA, USA), according to the manufacturer’s instructions, and quantified using a Nanodrop 2000 spectrophotometer (ThermoFisher Scientific, Monza, Italy). One μg RNA/sample was reversed-transcribed into cDNA using a RevertAid RT Reverse Transcription Kit (#00771469, ThermoFisher Scientific, Monza, Italy).

The analysis of gene expression was performed via real-time PCR using a Luna^®^ Universal qPCR Master Mix (#M3003E, New England Biolabs Inc., Ipswich, MA, USA), along with the forward and reverse primers (5 pmol each) reported in Table 1. The primer sets were designed according to the known sequences reported in GenBank with the Primer 3 program [25] (code available at http://www-genome.wi.mit.edu/genome_software/other/primer3.html, accessed on 5 March 2023). Real-time PCR was performed in a RotorGene 3000 apparatus (Corbett). A melting curve analysis was added at the end of each amplification cycle.

The data analysis was conducted according to the Pfaffl formula [26], and *RPL15* was used as housekeeping.
(1)$$Ratio=\frac{{Etag}^{∆CTtag(control-sample)}}{{Eref}^{∆CTref(control-sample)}}$$

### 2.7. Statistical Analysis

Graphical representation and statistical analysis were performed using GraphPad Prism 8.0 software (La Jolla, CA, USA).

Growth curve data were best fitted with the Gompertz growth model and analyzed using the extra sum-of-squares F test. Data are shown as mean ± SD. Asterisks indicate statistical significance (* *p* < 0.05; ** *p* < 0.01; *** *p* < 0.001).

Statistical differences in the intracellular iron quantity, and of real-time PCR data, among groups were determined using 2-way ANOVA, followed by Sidak’s multiple comparisons test. CTCF/area data were analyzed with 1-way ANOVA followed by Tukey’s multiple comparisons test.

Differences were considered significant when *p* < 0.05.

## 3. Results

### 3.1. Effect of Iron Conditioning on Cell Morphology

Via visual microscope analysis (Figure 1), HOG cells show an epithelioid-like and oligodendrocyte-like morphology; cells appear flat with long and thin extensions and a high nucleus/cytoplasm ratio.

### 3.2. Effect of Medium Composition on Cell Models

The iron content was evaluated in cells cultured in their medium and in cells for which the medium was changed for 24 h.

Intracellular iron, measured via ICP-MS, was significantly different between cells grown in 100 µM of ferric citrate compared to non-conditioned HOG cells (2,439 ± 722.3 fg/cell vs. 31.72 ± 7.59 fg/cell, *p* < 0.001), even in a medium without ferric citrate solution (Figure 2).

Interestingly, a 24 h medium switch induced a variation in the intracellular iron in both cell types. Intracellular iron decreased in HOG100 with the medium switched for 24 h without ferric citrate (1776.91 ± 576.17 fg/cell) but increased in control HOG cells with the medium switched for 24 h to 25 µM (122.51 ± 29.86, *p* < 0.01) or 100 µM ferric citrate-LG-DMEM (597.6 ± 83.8 fg/cell, *p* < 0.001).

### 3.3. Cell Line Viability and Proliferation after Conditioned Iron Medium

As 100 µM of sideremia is never detected clinically, we tested the effect of media with lower concentrations of iron to mimic the concentration of iron observed in patients with siderosis [27].

We monitored the iron-conditioned cells’ behavior (HOG 100 µM) regarding proliferation and growth in two different culture conditions (with and without ferric citrate 10 µM) versus non-conditioned HOG cells.

The growth curves generated by the Gompertz model reflect a classic growth curve pattern and were not different between siderotic (HOG100) and control cells, either grown in LG-DMEM- 1% FBS with or without 10 µM of ferric citrate, a concentration ten-fold higher than the CSF iron detected in transfused patients (HOG100 vs. HOG; *p* = 0.304) but lower than the serum levels (48.92 ± 12.97 µM).

Cell viability significantly changed when cells were switched from LG-DMEM to LG-DMEM plus 10 µM ferric citrate, showing an enhancement in the medium enriched with ferric citrate (*p* < 0.0001, Figure 3). Specifically, the exponential phase of both HOG and HOG 100 μM cultured in media supplemented with ferric citrate 10 μM lasted longer than the condition without medium supplementation (YM = 70,348 vs. 190,618, *p* < 0.0001), and the lag phase was shorter (K = 0.048 vs. 0.017; *p* < 0.05).

### 3.4. Confocal Microscopy Confirms the Iron Uptake

When HOGs were incubated with RhoNox-1, an active fluorescent probe that specifically detects unstable Fe^2+^, the increase in the Fe^2+^ molecules inside the cells incubated with high iron-containing media was observable mainly in mitochondria and cytoplasm (Figure 4 and Appendix A). We observed a significative increase in RhoNox-1 fluorescence after 24 h of switching to 10 µM Fe-citrate LG-DMEM. The analysis of co-localization with ImageJ [28] showed that only 1% of the Fe^2+^ was localized in the lysosomal compartment. Some of the RhoNox-1 fluorescence was also detected in the nuclei, suggesting a possible role of Fe^2+^ in the regulation of gene expression.

### 3.5. Gene Expression

To investigate the results of the Alamar Blue assay, we investigated the effects on the expression of representative genes related to cell growth and iron metabolism in HOG cells (Figure 5).

The expression of heavy and light chains of ferritin (*FTH1*, *FTL*) that organize the cell iron storage system did not change among all the tested conditions. On the contrary, *c-MYC* was significantly induced by media with elevated iron concentrations (100 µM Fe-citrate LG-DMEM 2.31 ± 0.45 folds; 10µM Fe-citrate LG-DMEM 2.49 ± 0.14 folds). The gene coding for the moonlighting protein glyceraldehyde 3-phosphate (*GAPDH*) was strongly induced using iron-enriched media (100 µM Fe-citrate LG-DMEM 4.42 ± 0.35 folds; 10µM Fe-citrate LG DMEM 3.74 ± 0.45 folds), and the 24 h switch from 100 µM Fe-citrate to 10 µM Fe-citrate LG-DMEM did not affect its expression. Tubulin beta 3 (*TUBB3*) expression was highly induced in the 100 µM Fe-citrate-enriched medium (4.42 ± 1.77 folds vs. LG-DMEM) but was normalized in 24 h when switched to 10 µM Fe-citrate medium (0.8 ± 0.046 folds).

## 4. Discussion

Previous studies have shown that HOG cells have a duplication time of about 18 h and a rather heterogeneous phenotype: they express some oligodendrocyte-specific proteins, including A2B5, a 15 kDa form of the basic protein of the myelin (MBP) and cyclic 2′,3′-nucleotide3′-phosphodiesterase (CNPase); even the lysosomal protein galactosylceramide (GalC), sulfogalactosylceramide, vimentin, and cytokeratin K7 were found to be expressed [23,29,30]. Data collected from Buntinx et al. show that HOG is a model of human oligodendrocytes ‘arrested’ in an immature developmental stage. Culturing in the appropriate medium can induce further differentiation of these cells [24].

Due to cell viability assays, it was possible to evaluate how HOG and HOG 100 µM growth can be influenced by the presence of iron in the cell culture medium. Generally, the classical growth curve pattern includes a lag phase after culture seeding, followed by a log phase in which cells proliferate and grow exponentially.

Finally, the stationary phase occurs when the growth medium is spent or when the cells occupy all the available substrates. The main difference in terms of growth is observed by comparing the presence or absence of iron in the medium, regardless of the preconditioning of the cells with iron. The exponential phase of both HOG and HOG 100 μM lasts longer, reaching the stationary phase almost at the end of the experiment, after two weeks of cell culture. We can hypothesize that the different conditions offered by the addition of the 10 μM ferric citrate solution may favor a more advantageous growth and proliferation condition in tumoral cells, as well as in healthy oligodendrocytes. It is known that because of their elevated metabolic needs associated with myelination, oligodendrocytes have the highest iron levels in the brain [31]. Brown and colleagues observed increased intercellular iron import in cancer cells [32].

Iron is a cofactor for several enzymes involved in the proliferation and differentiation of oligodendrocyte progenitor cells and in the production of cholesterol and phospholipids, which are essential myelin components [31]. In general, two major destinations of iron within mammalian cells are the cytosolic iron-storage protein named ferritin and mitochondria [33]. Ferritin is composed of heavy (Fth, FTH1) and light (Ftl, FTL) chains, capable of binding over [34]; it is the primary intracellular iron-storage protein and is essential for keeping iron in a soluble and nontoxic form [35]. The transferrin (TF)/transferrin receptor (TfR)/divalent metal transporter 1 (DMT1) pathway plays a significant role in iron transport in immature oligodendrocytes, and the proportion of iron transported by this pathway may decrease with the beginning of myelination [36,37]. Several proteins cooperate to maintain cellular iron homeostasis by regulating iron uptake and storage. TF is the most important iron carrier and has the capacity to bind two atoms of ferric iron with high efficiency. Tf loaded with iron (holo-Tf) binds with great affinity to the TfR localized on the cell surface, after which the Tf-TfR complex is internalized through receptor-mediated endocytosis. Ferric iron is then reduced to ferrous iron by metalloreductases and is transported across the endosomal membrane to the cytosol by DMT1. In the cytosol, iron may be transported by binding to chaperones that donate iron to specific target proteins, or it may traffic to the mitochondria [38].

It is known that iron promotes cell growth and proliferation, but it also causes oxidative stress damage. In particular, the aberrant accumulation of iron and subsequent excess ROS cause oxidative stress, which leads to genome and epigenome alterations, giving rise to tumor heterogeneity and evolving metastatic potential [32].

During viability assays, cells were also observed under the microscope and were viable, proliferating, and characterized by an oligodendrocyte-like morphology appearing flat with long and thin extensions and a high nucleus/cytoplasm ratio. Human oligodendroglial cell line HOG presents oligodendrocyte-associated features [23,24].

Confocal observations showed that the Rhonox-1 signal, linked to Fe^2+^, is mainly compartmentalized. It is strongly increased after a 24 h incubation in a medium containing 10 µM of Fe, a concentration detected in the serum of patients with systemic siderosis. Most of the signal is in the mitochondria and vesicles. Fe^2+^ is a form of iron that, through the Fantom reaction, induces the production of free radicals, leading to lipid peroxidation and ferroptosis [13]. During our observation, we did not detect any sign of apoptosis, meaning that, at least in HOG cells, the excess of Fe^2+^ is neutralized by mechanisms that require further investigation. Although 100 µM of extracellular iron is a condition that is never detected in vivo, our data support the importance of extracellular iron in the behavior of tumor cells. It is well known that cancer cells can release small extracellular vesicles that allow the transport of substances, among which is iron, from the cells themselves to communicate with their environment [39].

The HOG cell lines all exhibited markers of immature oligodendrocytes. Growing evidence suggests that iron-associated proteins, such as ferritin and the proto-oncogene *c-MYC*, contribute to the growth of malignant tumor cells. Ali and colleagues demonstrated that *FTH1* silencing promoted cell growth in breast cancer leading to an increased *c-MYC* expression and reduced cell sensitivity to chemotherapy [40,41]. Our data documented no change in *FTH1* and *FTL* gene expression, but *c-MYC* was significantly induced by media with elevated iron concentrations, supporting the higher proliferation rate observed in the HOG cell line.

We also observed a strong induction of *GAPDH* expression. Previous studies have revealed that GAPDH acts as a receptor for iron carrier proteins [42]. Iron is increasingly linked to cancer, such as neuroblastoma, because there is an elevated requirement for iron that favors cancer progression. However, it is unclear whether GAPDH-mediated iron uptake has any role in ferroptosis: cancer cells might utilize iron to favor their progression and to increase their invasive capability [42]. GAPDH localizes not only in the cytosol: this protein is detected also in the membrane, the nucleus, polysomes, the ER, and the Golgi. GAPDH has been shown to regulate microtubule assembling and vesicular trafficking from the ER to Golgi, the apparatus being involved in the early secretory pathway [43].

Our data show that iron-rich media prompt the expression of *TUBB3*; iron probably leads to increased message stability favoring gene transcription. *TUBB3* is frequently overexpressed in human tumors and is associated with resistance to microtubule-targeting agents, tumor aggressiveness, and poor patient outcomes. Together with tubulin, GAPDH could be part of a cellular response to the increased iron uptake the is remodeled by the reduction of the iron in the microenvironment.

## 5. Conclusions

In conclusion, this work provides new information on the great importance of iron in tumor development and progression. We observed that the extracellular iron concentration is more relevant than the intracellular one in the biology of tumor-derived cells. In particular, it is interesting to note the rapid variation in intracellular iron as the concentration of extracellular iron changes. As the metabolic and proliferative rates of tumor cells are usually higher than those of healthy cells, this leads to a substantial increase in iron demand and highlights the importance of iron in tumorigenesis and progression. An innovative therapeutic approach including the reduction of iron concentration in tumor cells could be used to bring about an antiproliferative effect to achieve better clinical results in pediatric OG patients.

## Figures and Tables

**Figure 1 biomedicines-11-02940-f001:**
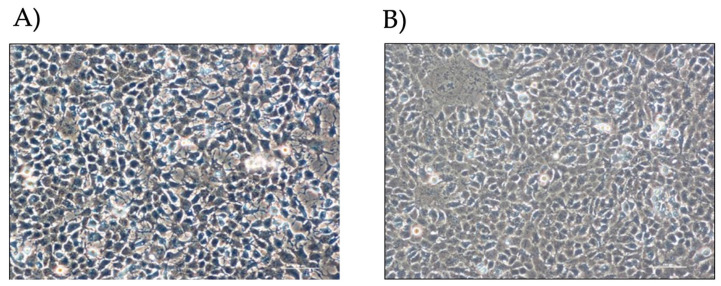
HOG cells under optical microscope. (**A**) HOG LG-DMEM, 1% FBS; (**B**) HOG LG-DMEM, 1% FBS, 100 µM Fe-citrate.

**Figure 2 biomedicines-11-02940-f002:**
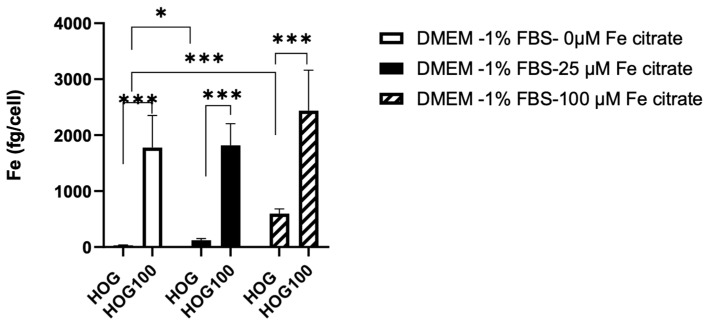
Effects of 24 h incubation of medium with different iron concentrations on the intracellular iron content of HOG cells. *p*-value according 2-way ANOVA (* *p* < 0.05; *** *p* < 0.001) followed by Sidak’s multiple comparisons test.

**Figure 3 biomedicines-11-02940-f003:**
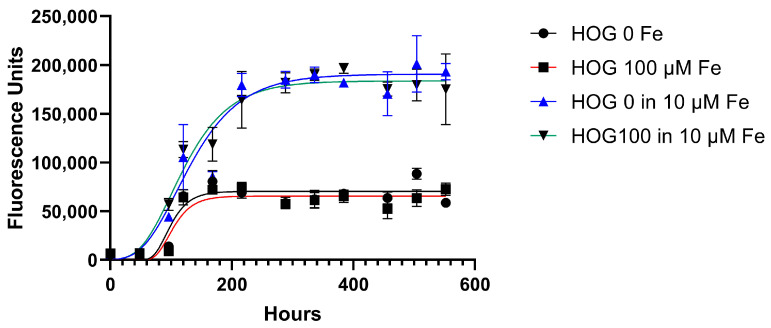
Growth curve of HOG and HOG 100 μM cell lines. Cells were cultured in a standard culture medium in the presence or absence of ferric citrate 10 µM. Cell viability was evaluated via resazurin metabolic conversion into living cells. Least squares regression was used, and data were best fitted by the Gompertz growth model. Data are expressed as mean ± standard deviation (SD) from at least three independent experiments for each cell line. Analysis was performed with GraphPad software 8.0 (H. J. Motulsky, GraphPad Curve Fit-ting Guide). http://www.graphpad.com/guides/prism/7/curve-fitting/index.htm (accessed on 2 February 2023).

**Figure 4 biomedicines-11-02940-f004:**
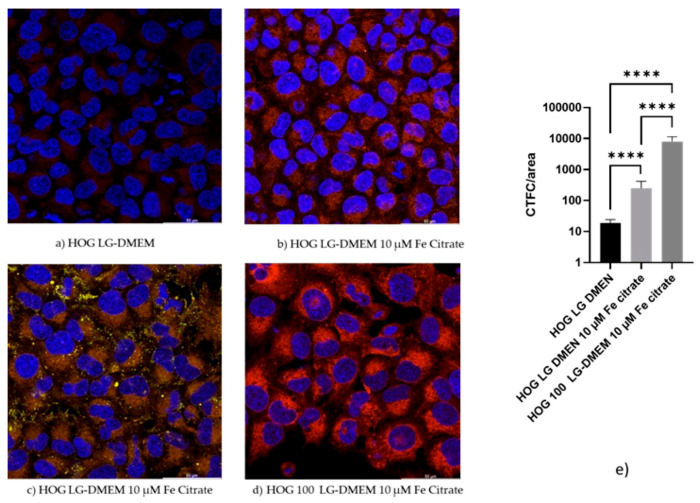
Confocal images of HOG and HOG100 cells. Cells were cultured in a standard culture medium in the presence or absence of ferric citrate 10 µM for 24 h. Thereafter, cells were incubated for one hour in a complete medium supplemented or not with 10 µM of ferric citrate, in the presence of 5 µM Rhonox-1 (red) and 1.6 µM LysoTracker^®^ Yellow HCK-123 (yellow). For nuclei counterstaining, 5 µg/mL of Hoechst 33-258 (blue) was added to all the tested conditions during the last 20 min of incubation. Images were collected with a Leica TCS SP5 Laser Scanning Confocal microscope (Leica Microsystems, Wetzlar, Germany). Fluorescence intensity of the single cells was acquired, and images were analyzed with ImageJ software. In picture (**a**), HOG cells in a low-iron-containing medium display a barely detectable Rhonox-1 signal. The red signal increases when cells are incubated in 10 µM of Fe-citrate, and it is much more evident in cells chronically incubated with 100 µM Fe-citrate LG-DMEN, although switched to 24 h in medium containing 10 µM Fe-citrate. In frame (**c**), yellow dots show the lysosomes, and some of them show an orange color due to the presence of Rhonox-1, which binds to Fe^2+^. (**e**) Graphical presentation of confocal images quantitation; fluorescence intensity of the single cells was acquired, and images were analyzed with ImageJ software ver 1.54g. *p*-value according 2-way ANOVA (**** *p* < 0.0001) followed by Sidak’s multiple comparisons test. CTCT: corrected total cell fluorescence.

**Figure 5 biomedicines-11-02940-f005:**
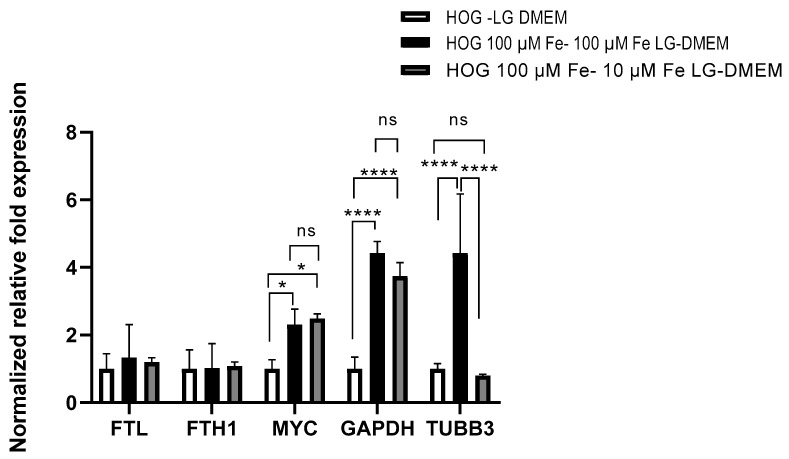
Gene expression analysis of HOG in LG-DMEM and HOG100 cultured in 100 µM Fe-citrate in LG-DMEM or switched for 24 h to LG DMEM + 10 µM Fe-citrate for 24 h. *p*-value according 2-way ANOVA (ns, not significative; * *p* < 0.05; **** *p* < 0.0001) followed by Sidak’s multiple comparisons test.

**Table 1 biomedicines-11-02940-t001:** Primer sequences (from https://primer3.ut.ee/ (accessed on 5 March 2023) for real-time PCR analysis.

Gene Name	Gene Bank Number	Primer Sequence (5′-3′)	Product Size (bp)	Annealing Temperature (°C)	Primer Efficiency
*GAPDH*	NM_002046	Forward TCTCTGCTCCTCCTGTTC Reverse GCCCAATACGACCAAATCC	120	60	1.08
*FTH1*	NM_002032	Forward ACCTGTCCATGTCTTACTACTTTG Reverse GCCACCTCGTTGGTTCTG	134	60	1.03
*FTL*	NM_000146	Forward GCCTCCTACACCTACCTCTC Reverse GCTGGCTTCTTGATGTCCTGG	179	60	1.16
*MYC*	NM_002467.4	Forward GCGACTCTGAGGAGGAACA Reverse TGCGTAGTTGTGCTGATGTG	182	59	0.96
*RPL15*	NM_002948	Forward GGAGAAAGAAGCAGTCTGATGTC Reverse CCACGGCGAACACGAATC	175	60	1.06
*TUBB3*	NM_006086	Forward GCAAGGTGCGTGAGGAGTAT Reverse GCGGAAGCAGATGTCGTAGA	182	60	1.00

## Data Availability

The original contributions presented in this communication are included in the article. Further inquiries can be directed to the corresponding author.

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
