# Peer review of "Iron Bioavailability in the Extracellular Environment Is More Relevant Than the Intracellular One in Viability and Gene Expression: A Lesson from Oligodendroglioma Cells"

_biomedicines, 2023, doi:10.3390/biomedicines11112940_

Round 1
Reviewer 1 Report
Comments and Suggestions for Authors
In this manuscript, Braidotti and colleagues assessed the role of iron in an in vitro model of human oligodendroglioma brain tumor cells by evaluating cell morphology, effects of the lateral yellow medium on cell growth, iron uptake, and iron metabolism-related gene expression by light microscopy, ICP-MS, confocal microscopy, and Real-time PCR. Overall, this paper is an interesting study. However, I still have some concerns about the current form of the manuscript. The authors need to address several aspects before this can be published as follows:
Main concerns:
1. The authors in the manuscript used different concentrations of ferric citrate in different experiments; on what basis did the authors determine the concentration of ferric citrate?
2. Since the authors are testing the effect of experimental iron on cell growth, it is necessary to provide growth curves of HOG cells under conditions where different concentrations of iron citrate are added to the medium.
3. The authors need to describe or discuss how iron ions enter HOG cells. are there membrane proteins in the HOG cell membrane that transport iron ions?
4. Although the authors noted a decrease in iron content within HOG 100 cells after 24 hours of incubation in an iron-free medium (Fig. 2), it was still much higher than the iron content of HOG cells in 25 or 100 µM iron citrate medium. Why did HOG 100 grow at the same rate as HOG in an iron-free medium (Fig. 3)? In addition, how does the iron content of HOG 100 change when cultured for a long time in an iron-free medium?
5. The authors studied gene expression under different conditions, so what about gene expression of HOG 100 under LG-DMEM culture conditions?
Comments on the Quality of English LanguageMinor editing of English language required
Author Response
Main concerns:
- The authors in the manuscript used different concentrations of ferric citrate in different experiments; on what basis did the authors determine the concentration of ferric citrate?
We thank the reviewer for the comment. We are aware that several iron compounds are known to be poorly soluble in PH 7.4 culture medium and with low serum concentrations. Iron citrate is completely soluble in DMEM with 1% FBS at 100 μM, so the medium was filtered with a 0.2 μm filter. Mediums at lower concentrations of iron citrate were prepared by diluting 100 μM of iron citrate medium.
- Since the authors are testing the effect of experimental iron on cell growth, it is necessary to provide growth curves of HOG cells under conditions where different concentrations of iron citrate are added to the medium.
We thank the reviewer for the clarification. For the evaluation of cell growth, curves with different concentrations have been performed at 0 and 100 µM, as shown in figure 3. The aim was to test whether the growth curve at 100 µM Fe in the medium was different in HOG 100 and control HOG since we did not observe differences in the control medium. We saw that there was no difference even in 100 µM medium, but the significative difference was due to the Fe concentrations in the medium. Demonstrating the toxicity of iron was not part of our hypothesis. However, at 100 µM, no toxicity for HOGs is observed, nor for all concentrations between 0 and 100 µM. We decided not to test higher concentrations because, in the human body, we cannot even reach 100 µM.
- The authors need to describe or discuss how iron ions enter HOG cells. Are there membrane proteins in the HOG cell membrane that transport iron ions?
According to the reviewer's indication, we have added in the manuscript (Discussion section, lines 301-319) details regarding the mechanism of iron transport into cells and the role of iron in oligodendrocytes.
We take this opportunity to specify that our goal was not to investigate the different mechanisms of iron entry into cells. Surely, investigating the entry mechanisms will be an important part of the project, but at this time, we can present only preliminary data about iron's effect on cells. Specifically, the aim of the preliminary study was to demonstrate the rapid variation in intracellular iron amounts with varying concentrations in the extracellular iron content. We confirmed the presence of iron-containing vesicles in the cells and observed the presence of iron in the nucleus and mitochondria for the first time with induction of gene expression at 24 hours.
- Although the authors noted a decrease in iron content within HOG 100 cells after 24 hours of incubation in an iron-free medium (Fig. 2), it was still much higher than the iron content of HOG cells in 25 or 100 µM iron citrate medium. Why did HOG 100 grow at the same rate as HOG in an iron-free medium (Fig. 3)? In addition, how does the iron content of HOG 100 change when cultured for a long time in an iron-free medium?
We thank the reviewer for the comment. The rapid increment of extracellular iron concentration is associated with a rapid increase of intracellular iron; the free ions are quickly removed from the cytoplasm when stored in vesicles. Otherwise, the release of the stored iron appears to proceed slowly. In Figure 2, the decrease in iron content within HOG 100 cells after 24 hours of incubation in an iron-free medium is observed.
The authors hypothesized the same growth rate between HOG100 and HOG in a medium with/without iron due to the presence of iron out of cells and the entry of iron into the cell and, therefore, into the mitochondria. In the future, we think it will be necessary to assess the energy status of the cell e.g., to measure the ATP levels. However, the iron present in HOG100 is largely sequestered in vesicles and, therefore, does not participate in cellular metabolism.
- The authors studied gene expression under different conditions, so what about gene expression of HOG 100 under LG-DMEM culture conditions?
The reviewer noted that we did not evaluate in HOG 100 without iron in the culture medium the expression of representative genes related to cell growth and iron metabolism. From the results obtained in the growth curves (Figure 2), we have observed that the iron concentration of the medium influences growth. Since HOG 0 and HOG 100 cells behave similarly in growth curves, with equal iron within the medium, we thought it would have been a duplication of control. It was interesting for us to consider HOG 100 in a medium with 10 µM of iron that is easily found in siderotic patients, and we wanted to check if lowering iron centration equally affects the induction of gene expression observed in HOG100 We hope that the reviewer will also agree with us on this point.
Reviewer 2 Report
Comments and Suggestions for Authors
The article presented by Braidotti et al. focuses on developing an in vitro model of oligodendroglioma cells under chronic exposure to iron concentrations, aiming to simulate a siderotic condition in tumor cells. I find the idea interesting, as iron metabolism in gliomas is not yet well understood, and the effect of iron at various cellular levels could be exploited to develop therapies or trigger cell death through ferroptosis. However, I have several doubts about the experimental design and results obtained that need clarification to understand the potential of the article:
-
In the introduction, there is mention of iron passing through the BBB and speculation about its passage and the possible massive exposure to iron when therapies are applied to tumor cells. This statement is not clear and appears overly speculative without experimental or bibliographic support.
-
Ferritin and its possible role in intracellular iron accumulation are mentioned, but the interpretation of this protein, its plasma level, and its levels in tumor tissue are not clear. The authors make some statements about this without providing bibliographic references to support these associations or their significance in this article since ferritin levels are not measured in the developed model.
-
In this regard, based on what experience or reference was the decision made to use the condition of 100 μM of ferric citrate to develop the model? Was cell viability evaluated under this condition? The article also does not explain why exposure was later reduced to 10 μM or 25 μM in some assays. Are these levels appropriate, do they induce oxidative stress or ferroptosis? These events should have been observed and reported to better understand the proposed model.
-
In the quantification of iron by ICP-MS in cell lysates, how was the quantification normalized per cell? The methodology does not mention this, and the results are later expressed in this unit. Normalization to protein quantification might be more appropriate.
-
In the confocal microscopy experiments, the images need a more detailed description of the fluorescence they are showing. The article also does not explain how colocalization analysis was conducted. Was CTC used for this analysis? If so, this is not correct. There are more appropriate colocalization index analyses to apply.
-
In the gene expression assay, there seems to be a missing comparison with the HOG-LG 100 μM cell group exposed to DMEM as a control or treatment for the developed model.
-
Measuring ferritin protein levels should be a priority to explain the relationships with iron accumulation.
-
The following argumentation in the discussion is highly speculative and requires experimental support: "Fe2+ is a form of iron that through the Fantom reaction induces the production of free radicals leading to lipid peroxidation and ferroptosis. During our observation, we did not detect any sign of apoptosis, meaning that at least in HOG cells, the excess of Fe2+ is neutralized by mechanisms that require further investigation. Although 100 μM of extracellular iron is a condition that is never detected in vivo, our data support the importance of extracellular iron in the behavior of tumor cells. It is well known that cancer cells could release small extracellular vesicles that allow the transport of substances, among which iron, from the cells themselves to communicate with their environment [26]. The removal of intracellular iron by vesicles is an important mechanism driving ferroptosis and so tumor progression."
Author Response
- In the introduction, there is mention of iron passing through the BBB and speculation about its passage and the possible massive exposure to iron when therapies are applied to tumor cells. This statement is not clear and appears overly speculative without experimental or bibliographic support.
We thank the reviewer for the comment, and according to his/her suggestion, we have added some details in the introduction of the manuscript (PMID: 35530363). The role of side effects of chemoradiotherapy on the BBB is speculated. As mentioned in lines 68-70 to date, “It is unclear whether tumor resistance to conventional therapies occurs because of increased permeability of the BBB due to conventional pharmacological therapies or radio-therapy, allowing massive entry of iron without control.” This is certainly an aspect that is mainly interesting for our study, so we will try to explore it in greater depth in the future when we have the opportunity to work with the BBB model.
However, iron crosses the BBB, as well as many other molecules. In literature, the existence of a transport system that transfers iron into the BBB is currently known (PMID: 25355056, PMID: 2623618, PMID: 19386095). In addition, it is known that cells receive iron support for growth and proliferation (PMID: 12242109), especially cancer cells (PMID: 21324793). Inducing iron depletion could be a therapeutic advantage to manage siderosis. Iron chelators have long been known for their effects on iron reduction and antiproliferative effects (PMID:7492790, PMID: 11468187). With respect to the theme of resistance, chemo-radio therapies applied could damage the BBB. Therefore, the iron entry could be altered, and that would establish a mechanism that favors tumor growth. Liu et al. 2013 discussed the excessive iron role in brain injury (PMID: 23079850).
- Ferritin and its possible role in intracellular iron accumulation are mentioned, but the interpretation of this protein, its plasma level, and its levels in tumor tissue are not clear. The authors make some statements about this without providing bibliographic references to support these associations or their significance in this article since ferritin levels are not measured in the developed model.
We thank the reviewer for this observation. We previously mentioned ferritin in the introduction section of the manuscript, briefly explaining its role in iron accumulation. Some details about ferritin and its role are now added to the discussion (lines 310-312). Ferritin has long been established as an iron storage protein with a central role in iron regulation, even in the brain, in which the oligodendrocytes, microglia, and neurons express ferritin. Moreover, studies have reported that ferritin (H form) is taken up into the brain parenchyma and its trafficking across the BBB via endothelial cells, as previously mentioned. Elevated serum ferritin levels are observed in different types of cancers (PMID: 37209958), but the relationship between ferritin and glioma is poorly understood yet. High serum ferritin levels are reported in numerous cancers, including glioblastoma, neuroblastoma, breast, renal, and cervical cancers, and are often associated with poor survival of these patients. The dosage of ferritin in patients is part of clinical routine measurements, as well as the evaluation of iron accumulation in the organs using other methods (MRI to evaluate multiorgan siderosis). The role of ferritin is not central in our preliminary study, and no dosages have been pursued in the model. However, we observed no changes in our cell model's light or heavy ferritin chain expression. This could indicate that ferritin synthesis in HOGs does not contribute to the control of accumulated iron in the HOG model.
- Based on what experience or reference was the decision made to use the condition of 100 μM of ferric citrate to develop the model? Was cell viability evaluated under this condition? The article also does not explain why exposure was later reduced to 10 μM or 25 μM in some assays. Are these levels appropriate, do they induce oxidative stress or ferroptosis? These events should have been observed and reported to better understand the proposed model.
We thank the reviewer for clarifying and apologize for these aspects being unclear. We think we have already answered these questions posed by reviewer 1 (question 1). We are aware that several iron compounds are known to be poorly soluble in PH 7.4 culture medium and with low serum concentrations. Iron citrate is completely soluble in DMEM with 1% FBS at 100 μM, so the medium was filtered with a 0.2 μm filter. Mediums at lower concentrations of iron citrate were prepared by diluting 100 μM of iron citrate medium.
However, HOGs did not show signs of apoptosis, so they appear resistant to siderosis, thanks to the inclusion of iron in vesicles. In our preliminary study, the expression of anti-radical mechanisms has not yet been evaluated, but it will be one of the aspects to be considered in future analysis.
- In the quantification of iron by ICP-MS in cell lysates, how was the quantification normalized per cell? The methodology does not mention this, and the results are later expressed in this unit. Normalization to protein quantification might be more appropriate.
According to other authors, we decided to refer to the quantity of iron as the number of cells and not to that of proteins (e.g., Shashank et al. 2018, PMID: 28888202). Cells were counted and then fixed in paraformaldehyde. Measuring proteins in ICP-MS would have allowed iron to leak out of cells, introducing errors. Furthermore, variations in culture conditions could induce a difference in the intracellular amount of proteins, which we have not currently evaluated.
- In the confocal microscopy experiments, the images need a more detailed description of the fluorescence they are showing. The article also does not explain how colocalization analysis was conducted. Was CTC used for this analysis? If so, this is not correct. There are more appropriate colocalization index analyses to apply.
We implemented the reviewer's advice and improved the figure description by adding details about the observed fluorescence.
According to Bolte, S., & Cordelières, F. P. (2006). We have analyzed colocalization by a one to one pixel matching analysis using JACOP plug-in of Image-J (PMID: 17210054).
- In the gene expression assay, there seems to be a missing comparison with the HOG-LG 100 μM cell group exposed to DMEM as a control or treatment for the developed model.
We thank the reviewer for raising the point. For further clarity, we modified Figure 5, highlighting the comparison between control and treatment. The description of the figure has also been improved.
- Measuring ferritin protein levels should be a priority to explain the relationships with iron accumulation.
We thank the reviewer for the observation. We mention ferritin, explaining its role in iron accumulation. We intend to deepen this focus in the future. However, as previously mentioned, the role of ferritin is not central to our preliminary study, and no dosages have been pursued in the model. As a preclinical study, we evaluated only heavy and light chains of ferritin (FTH1, FTL) as gene expression in this cell model, and it didn’t change.
- The following argumentation in the discussion is highly speculative and requires experimental support: "Fe2+ is a form of iron that through the Fantom reaction induces the production of free radicals leading to lipid peroxidation and ferroptosis. During our observation, we did not detect any sign of apoptosis, meaning that at least in HOG cells, the excess of Fe2+ is neutralized by mechanisms that require further investigation. Although 100 μM of extracellular iron is a condition that is never detected in vivo, our data support the importance of extracellular iron in the behavior of tumor cells. It is well known that cancer cells could release small extracellular vesicles that allow the transport of substances, among which iron, from the cells themselves to communicate with their environment [26]. The removal of intracellular iron by vesicles is an important mechanism driving ferroptosis and so tumor progression."
We thank the reviewer for raising the point. As specified in this part of the discussion, we are making speculations, which we will clarify in the future with new experiments. The work presented here is, in fact, preliminary, and we are aware that we need to carry out more in-depth investigations. We have implemented the following references in the main text:
à"Fe2+ is a form of iron that, through the Fantom reaction, induces the production of free radicals leading to lipid peroxidation and ferroptosis PMID: 20431983
Reviewer 3 Report
Comments and Suggestions for Authors
The development of an “in vitro” model for a neoplasm is interesting and relevant. However, some points should be clarified:
1. I am not sure about the pertinence in the title of …”as a useful tool to study iron role in tumor cell”. Which would limit the potential use of the model for other research (e.g. genomic studies).
2. The authors should clarify the novelty of the method of “in vitro” oligodendroglioma, as at least a cell line of the tumor exists in the market (from Sigma Aldrich).
3. If this text is not the presentation of a new experimental model, but a study of iron metabolism on an already developed experimental model it should be explicit.
Comments on the Quality of English LanguageNo comments.
Author Response
- I am not sure about the pertinence in the title of …”as a useful tool to study iron role in tumor cell”. Which would limit the potential use of the model for other research (e.g. genomic studies).
We thank the reviewer for the suggestion. We changed the title to “Iron bioavailability in the extracellular environment is more relevant than the intracellular one on viability and gene expression, a lesson from oligodendroglioma cells.”
- The authors should clarify the novelty of the method of “in vitro” oligodendroglioma, as at least a cell line of the tumor exists in the market (from Sigma Aldrich).
We thank the reviewer for the observation. The novelty does not lie in the cellular type of oligodendroglioma since there are several immortalized models like this (as the reviewer has pointed out). The novelty for us lies in a model that recapitulates brain cell iron overload, which can play a role in the proliferation and growth of brain cancer cells. Data presented in the manuscript are preliminary. In fact, the HOG and HOG100 models will be useful for us in future studies on the BBB. The long-term aim will be to clarify the causes of the massive permeability of iron through the dysfunctional BBB, which can determine the establishment of a pro-tumoral microenvironment and understand the mechanisms that may lead to increased iron permeability across the BBB in patients. In addition, the siderotic cellular model will also be useful for other purposes. Therefore, it is not relevant for the authors to specify right now for which future applications it will be used.
- If this text is not the presentation of a new experimental model, but a study of iron metabolism on an already developed experimental model it should be explicit.
We thank the reviewer for the comment. We think it can be both. The manuscript presents HOG100 as a siderotic cellular model, and preliminarily, results investigate the role of iron in the siderotic cellular model (HOG100) compared to control (HOG), i.e., how it is internalized and how it is exploited for cellular growth. For this reason, several assays were performed with different concentrations of iron in the culture medium.
Round 2
Reviewer 1 Report
Comments and Suggestions for Authors
The metabolic and proliferative rate of tumor cells is usually higher than that of healthy cells, leading to a substantial increase in the demand for iron. This highlights the importance of iron in tumorigenesis and progression. In this manuscript, cell morphology, the effect of iron-containing media on cell growth, iron uptake, and the expression of genes related to iron metabolism were assessed. In the updated manuscript, the authors made improvements. These revisions basically answered my concerns. I have no more questions for the authors to answer.
Author Response
We thank the reviewer for his comments.
Reviewer 2 Report
Comments and Suggestions for Authors
I don't think the new title provided is the most appropriate to describe the article. I believe the authors continue to make many speculative claims without experimental support and should focus on providing a more detailed description of the results. I consider the article to be preliminary. Many players in iron metabolism could have been measured in the developed model. It's not clear why the chosen concentration was used. Is it meant to mimic a specific condition in patients with this type of tumors?
Please provide a conclusion that aligns better with the main findings of the study and any new potential that may arise from this work.
Author Response
I don't think the new title provided is the most appropriate to describe the article. I consider the article to be preliminary. Many players in iron metabolism could have been measured in the developed model.
We thank the reviewer for his comments.
The new title, “Iron bioavailability in the extracellular environment is more relevant than the intracellular one on viability and gene expression, a lesson from oligodendroglioma cells.” is intended to be more focused on the aim of this study even though the study presented as communication is preliminary. An in-depth assessment of iron management by HOG cells is required to describe this model. This was not done because it was not within the aim of the present communication.
It's not clear why the chosen concentration was used. Is it meant to mimic a specific condition in patients with this type of tumor?
This is a significant comment.
For the evaluation of cell growth, curves with different concentrations have been performed at 0 and 100 µM, as shown in Figure 3. We aimed to test whether the growth curve at 100 µM Fe in the medium differed in HOG 100 and control HOG since we did not observe differences in the control medium. We saw that there was no difference even in 100 µM medium, but the significative difference was due to the Fe concentrations in the medium. Demonstrating the toxicity of iron was not part of our hypothesis. However, at 100 µM, no toxicity for HOGs is observed, nor for all concentrations between 0 and 100 µM. We decided not to test higher concentrations because, in the human body, we cannot even reach 100 µM.
On the contrary, concentrations of 10 and 25 µM are found by measuring sideremia in patients with siderosis. We wanted to show how the concentration of iron in the extracellular milieu is more relevant than that of the intracellular one in the biology of cells derived from this tumor. Intriguingly, we have observed the rapid variation in intracellular iron as the concentration of extracellular iron changes. Moreover, tumor cells' metabolic and proliferative rate is usually higher than healthy cells, leading to a substantial increase in iron demand. This highlights the importance of iron in tumorigenesis and progression. In this manuscript, we assessed cell morphology, the effect of iron-containing media on cell growth, iron uptake, and the expression of genes related to iron metabolism. As suggested, this developed model will measure other players in iron metabolism.
To improve the comprehension of the reasons for the iron concentrations used, we added a sentence: As 100 µM of sideremia is never detected clinically, we tested the effect of media with lower concentrations of iron to mimic the concentration of iron observed in patients with siderosis (Results, line 238-240).
Please provide a conclusion that aligns better with the main findings of the study and any new potential that may arise from this work
Thank you very much for your comment
The conclusion has been improved as follows: “In conclusion, this work provides new information on the great importance of iron for tumor development and progression. We have observed that the extracellular iron concentration is more relevant than the intracellular one in the biology of tumor-derived cells. In particular, it is interesting to note the rapid variation of intracellular iron as the concentration of extracellular iron changes. As the metabolic and proliferative rate of tumor cells is usually higher than that of healthy cells, this leads to a substantial increase in iron demand and highlights the importance of iron in tumorigenesis and progression. An innovative therapeutic approach, including reducing iron concentration in tumor cells, could produce an antiproliferative effect to achieve better clinical results in pediatric OG patients.”
Round 3
Reviewer 2 Report
Comments and Suggestions for Authors
I have no more comments. I encourage the authors in the future to continue with deeper studies with the proposed cellular model.